# Participant Choice towards Receiving Potential Additional Findings in an Australian Nephrology Research Genomics Study

**DOI:** 10.3390/genes13101804

**Published:** 2022-10-06

**Authors:** Rosie O’Shea, Alasdair Wood, Chirag Patel, Hugh J. McCarthy, Amali Mallawaarachchi, Catherine Quinlan, Cas Simons, Zornitza Stark, Andrew J. Mallett

**Affiliations:** 1KidGen Collaborative, Australian Genomics Health Alliance, Murdoch Childrens Research Institute, Melbourne, VIC 3052, Australia; 2Genetic Health Queensland, Royal Brisbane & Women’s Hospital, Brisbane, QLD 4029, Australia; 3Departments of Nephrology, Sydney Children’s Hospitals Network, Sydney, NSW 2031, Australia; 4Department of Clinical Genetics, Royal Prince Alfred Hospital, Sydney, NSW 2050, Australia; 5Department of Nephrology, Royal Children’s Hospital, Melbourne, VIC 3052, Australia; 6Department of Paediatrics, Royal Children’s Hospital, Melbourne, VIC 3052, Australia; 7Faculty of Medicine, University of Melbourne, Melbourne, VIC 3052, Australia; 8Murdoch Children’s Research Institute, Melbourne, VIC 3052, Australia; 9Victorian Clinical Genetics Service, Murdoch Children’s Research Institute, Melbourne, VIC 3052, Australia; 10Department of Renal Medicine, Townsville University Hospital, Townsville, QLD 4814, Australia; 11Institute for Molecular Bioscience, The University of Queensland, Brisbane, QLD 4072, Australia; 12College of Medicine & Dentistry, James Cook University, Townsville, QLD 4814, Australia

**Keywords:** kidney disease, genomic testing, additional findings, choice

## Abstract

The choices of participants in nephrology research genomics studies about receiving additional findings (AFs) are unclear as are participant factors that might influence those choices. ***Methods***: Participant choices and factors potentially impacting decisions about AFs were examined in an Australian study applying research genomic testing following uninformative diagnostic genetic testing for suspected monogenic kidney disease. ***Results***: 93% of participants (195/210) chose to receive potential AFs. There were no statistically significant differences between those consenting to receive AFs or not in terms of gender (*p* = 0.97), median age (*p* = 0.56), being personally affected by the inherited kidney disease of interest (*p* = 0.38), or by the inheritance pattern (*p* = 0.12–0.19). Participants were more likely to choose not to receive AFs if the family proband presented in adulthood (*p* = 0.01), if there was family history of another genetic disorder (*p* = 0.01), and where the consent process was undertaken by an adult nephrologist (*p* = 0.01). ***Conclusion***: The majority of participants in this nephrology research genomics study chose to receive potential AFs. Younger age of the family proband, family history of an alternate genetic disorder, and consenting by some multidisciplinary team members might impact upon participant choices.

## 1. Introduction

The use of genomic sequencing in routine clinical care allows for vastly improved diagnostic yields for genetic kidney disease [1,2]. The yield is in the order of 30–60% combining adult and paediatric contexts [1,2,3]. Genomic sequencing technology benefits patients in a clinical diagnostic context through accurate clinical diagnosis, informing management, and familial implications [1,2]. In one Australian study, 39% of kidney disease patients had a change in their clinical diagnosis, with 56% having a change to their clinical management, such as: 13% avoiding the need for diagnostic renal biopsy, 44% changing surveillance, and 20% changing the treatment plan. Importantly, cascade testing was offered to 50% of families and 79% had an impact on the management of family members [1].

However, the vast amount of genomic data output from testing brings to the forefront the option for patients to receive additional findings (AFs) which might be actively sought or incidentally identified. The definition of AFs is “the results of a deliberate search for pathogenic or likely pathogenic variants in genes that are not apparently relevant to a diagnostic indication for which the sequencing test was ordered” [4]. The potential for AFs varies depending upon the type of genomic testing being undertaken across targeted gene panels, whole exome sequencing (WES), whole genome sequencing (WGS) or virtual gene panels applied to WES or WGS. In 2013, the American College of Medical Genetics (ACMG) produced policy guidelines recommending a defined set of actionable genes to be reported on from clinical exome and genome sequencing [4]. The set of 59 genes include hereditary cancer and cardiac genetic conditions where management recommendations are well established [4,5,6], and this list has recently been updated to include 70 genes [6]. The ACMG actionable list includes genes that are associated with conditions relevant to kidney medicine such as *KCNQ1* pathogenic variants causative for long QT syndrome [4,5,6] for which pre transplant defibrillator implantation in kidney transplant recipients would be considered if the presence of such a causative variant was known. 

Informed genomic consent requires an ethical lens to ensure the principles of autonomy, justice, beneficence, and non-maleficence facilitate patient choice to receive AFs in a shared decision-making model of patient centred care [7]. A single site United States (US) qualitative examination of 37 cancer and diagnostic odyssey patients’ preference to be informed about AFs revealed a diverse set of attitudes toward AF receipt [8]. Attitudes ranged from wanting to obtain all information about health risks to; alleviate uncertainty, provision of familial information, engage in preventative measures, and ensure quality of life, to not wanting to receive extra findings due to; religious or personal beliefs, burden of information, anxiety and worry or concerns. These findings in regard to disclosure of potential AFs are further validated by larger scale diagnostic genomic studies in both general [9] and cancer genetics [10] settings. The diverse set of views shows the importance of patient choice to be a core genomic consent component of the patient clinician interaction for clinical genomic sequencing [8]. Additionally, public opinions gathered through surveys of consumer preferences and decision making in Canada placed an important value on having a choice about which AFs they would receive. Choice of AF receipt depended on available information about high-penetrance, treatable disorders or high penetrance disorders with or without available treatment [11].

Genomic management in kidney disease requires a considered approach to AFs from clinicians, scientists, patients, and participants. Preferences regarding AFs by Australian nephrology research genomics participants are unclear. Our study aimed to describe Australian participant choices to receiving potential AFs as part of participation in a nephrology research genomics project.

## 2. Materials and Methods

The protocol and methods for this research genomics study have been previously published [12] with participants undergoing research WES and/or WGS in trios or extended pedigrees. In this study, AFs were not actively sought or identified as part of the study analysis or participation, and consent was sought only as to the potential disclosure of an AF or AFs if they happened to be identified as part of the genomic analysis in seeking a genetic aetiology for the relevant family’s or participant’s kidney phenotype and disorder. 

A retrospective review of consent choices about AFs amongst research genomics participants was undertaken within this national study of diagnostically refractory inherited kidney disease (May 2014–December 2020). Statistical analyses were undertaken using Prism (V9.2.0, GraphPad Software, San Diego, CA, USA) with results expressed as frequencies and percentages for categorical variables, and median (IQR) for continuous non-normally distributed variables. Relative risk, Mann–Whitney U-Test, Wilcoxon, and T-Test analyses were undertaken to assess relationships between variables and choice of consenting to AFs.

## 3. Results

In a cohort of 210 participants who underwent research genomic sequencing and analysis following uninformative diagnostic genetic testing for suspected monogenic kidney disease, 93% (195/210) chose to receive AFs. A minority (15/210; 7%) chose not to receive AFs. The study protocol did not include systematic analysis for AFs, and no AFs were returned as part of the study. The results describe participant preferences regarding receiving AFs, elicited as part of the consent process. The retrospective review of genomic consent of the 210 participants from 55 unrelated families revealed some participant characteristic differences in regard to choice to receive AFs.

There were no statistically significant differences between those consenting to receive AFs or not in terms of gender (male 47% vs. 47%, *p* = 0.97), median age (37.43 years vs. 43.92 years, *p* = 0.56), being personally affected by the inherited kidney disease of interest (50% vs. 60%, *p* = 0.38), or by the inheritance pattern (*p* = 0.12–0.19) (Table 1).

There were however statistically significant differences in consenting to not receive AFs and the family proband presenting in adulthood (*p* = 0.01), having family history of another genetic disorder (*p* = 0.01), and where the consent process was undertaken by an adult nephrologist (*p* = 0.01).

## 4. Discussion

The majority of Australian nephrology research participants are likely to consent to receive potential AFs. Familial factors and experience of inherited conditions in families may influence decisions in those choosing not to receive AF. A considered patient-centred approach to informed genomic consent for AF is required.

The initial ACMG policy statement and guidelines recommended all AFs to be returned to patients without giving patients explicit choice in the matter [4]. The active search for variants in genes recommended by the ACMG led to international debate with a revision to the policy statement allowing patients the opportunity to opt out of the analysis of medically actionable genes [13]. Outside of the US landscape a measured response to returning and consenting AFs in Europe and Australia—in the clinical and research domain—advocates for patient choice promoting a focus on dynamic consent between patients, clinicians, and laboratories [14,15,16,17]. In the Australian nephrology context, our study results indicate the importance of choice in the genomic consent process as 7% of participants chose not to receive AFs. There is limited knowledge about potential participant choices regarding AFs as part of research genomic studies in nephrology. For context, a prospective cohort of chronic kidney disease patients as part of a networked biobank in the United States found that 96% consented to research genetic studies at their initial study visit, with 4% declining. A dynamic nature of genomic consent at subsequent follow up visits did not change the overall acceptance or declining rate to consent [18]. These findings indicate that the majority of kidney disease patients are willing to engage in genomic research and receive AFs but researchers, clinicians, and laboratories need to consider that a minority will choose not to.

We noted with interest in our study that in addition to familial factors and experience of inherited conditions there appeared to be differences in consenting towards potential AFs based upon the type of clinician undertaking the consenting process. The vast majority of instances where a participant chose not to receive potential AFs were when the consent was undertaken by an adult nephrologist. Conversely, there were no such instances when the consent was undertaken by a pediatric nephrologist. Given that an adult nephrologist is more likely to encounter and undertake consent for adult patients this aligns with our parallel finding that the substantial majority of instances of a participant not consenting to potential AFs were when the proband in that participant’s family had presented in adulthood. Thus, we hypothesize that the age of a participant and that of the proband within their family may have an influence upon preferences towards receiving potential AFs. We cannot however exclude that there is not an influence or interaction of such AF consenting preferences with the type of clinician undertaking that consent, and we suggest that this should be further explored in additional and larger studies in other jurisdictions. 

A real-world example of the return of AFs to 104 prospectively recruited nephrology research participants identified AFs being reported in 8% of individuals amongst the then 59 ACMG medically actionable secondary genes. Within a US clinical care setting, 62% of participants were able to be re-contacted and results returned to 39% [19]. For those where a kidney genetic diagnosis was made, this had direct implications for patients’ nephrology care, such as; implications for therapy (54%), informed clinical prognosis (71%), and referrals for workup of associated extra kidney manifestations (66%) [19]. Some AFs reported had an impact on the patient’s kidney care as a pathogenic variant in *SCNA5* for Long QT type 3 required avoidance of medications that prolong the QT interval or deplete serum magnesium and potassium levels to mitigate the risk of sudden death. A return of results of clinical workflow was implemented and highlighted 20 key challenges indicating the importance of translational and implementation science research to inform a renal health services approach to genomic consent and return of results.

Genomic consent in nephrology is becoming a routine part of care. Understanding patient preference for the return of AF results allows clinicians to undertake a nuanced approach to consent conversations. However, when moving from the research setting into the real-world clinical context the health system’s capacity to undertake systematic screening for AFs may not be feasible without additional infrastructure, resources, and adaptations to service delivery [20,21]. The challenges for the system, services, and patients include the following; the additional cost, time for laboratory analysis, human power, resources, and education needed to understand management of AFs, especially when there is no family history of the genetic disease [20,21]. Therefore, a considered, patient-centred approach to clinical genomic consent in nephrology care is needed especially as we transition from research to predominantly clinical testing.

## Figures and Tables

**Table 1 genes-13-01804-t001:** Characteristics of the Additional Findings (AF) choice groups.

	Participant Choice towards AF	AF “Yes” vs. AF “No”	AF “No” vs. AF “Yes”
Variable	Yes (% of All)	No (% of All)	Mann–Whitney U Test Z Score, U, *p* Value	Chi^2^, *p* Value	Relative Risk (95%CI, *p* Value)	Relative Risk (95%CI, *p* Value)
** Total **	195 (93)	15 (7)	
**Median Age (years)**	37.43	43.92	1328.5, −0.59, *p* = 0.56	
**Consanguinity**	29 (15)	1 (7)		0.77, *p* = 0.38	1.05 (0.97–1.13, *p* = 0.24)	0.43 (0.06–3.14, *p* = 0.40)

** *Male* **	92 (47)	7 (47)		0.0015, *p* = 0.97	1.00 (0.93–1.08, *p* = 0.97)	0.98 (0.37–2.61, *p* = 0.97)
** *Female* **	103 (53)	8 (53)	1.00 (0.93–1.08, *p* = 0.97)	1.02 (0.38–2.71, *p* = 0.97)
** Proband Age in Family **
** *Paediatric Proband* **	106 (54)	3 (20)		6.59, *p* = 0.01	1.10 (1.02–1.19, *p* = 0.01)	0.23 (0.07–0.80, *p* = 0.02)
** *Adult Proband* **	89 (46)	12 (80)	0.91 (0.84–0.98, *p* = 0.01)	4.32 (1.25–14.86, *p* = 0.02)
** Family History of other genetic disorder **
** *Positive FHx* **	21 (11)	5 (33)		6.54, *p* = 0.01	0.85 (0.71–1.03, *p* = 0.11)	3.54 (1.31–9.54, *p* = 0.01)
** *Negative FHx* **	174 (89)	10 (67)	1.17 (0.97–1.42, *p* = 0.11)	0.28 (0.10–0.76, *p* = 0.01)
** Inheritance pattern **
** *Autosomal Recessive (AR)* **	124 (64)	7 (47)		1.70, *p* = 0.19	1.05 (0.97–1.15, *p* = 0.23)	0.53 (0.20–1.40, *p* = 0.20)
** *Autosomal Dominant (AD)* **	65 (33)	8 (53)	2.46, *p* = 0.12	0.94 (0.86–1.03, *p* = 0.16)	2.15 (0.81–5.68, *p* = 0.12)
** *Mitochondrial* **	6 (3)	0 (0)	N/A	1.08 (1.04–1.12, *p* = 0.0001)	0.95 (0.06–14.24, *p* = 0.97)
** Consented by **
** *Adult Nephrologist* **	101 (52)	13 (87)		6.83, *p* = 0.01	0.90 (0.84–0.94, *p* = 0.01)	5.47 (1.27–23.66, *p* = 0.02)
** *Paediatric Nephrologist* **	58 (30)	0 (0)	N/A	1.11 (1.05–1.17, *p* = 0.0001)	0.08 (0.01–1.38, *p* = 0.08)
** *Clinical Geneticist* **	17 (9)	2 (13)	0.36, *p* = 0.55	0.96 (0.82–1.13, *p* = 0.62)	1.55 (0.38–6.35, *p* = 0.55)
** *Genetic Counsellor* **	19 (10)	0 (0)	N/A	1.09 (1.04–1.13, *p* = 0.0001)	0.31 (0.02–4.98, *p* = 0.41)
** Relationship to affected family member/s **
** *Self* **	98 (50)	9 (60)		0.53, *p* = 0.47	0.97 (0.90–1.05, *p* = 0.47)	1.44 (0.53–3.91, *p* = 0.47)
** *Relative* **	97 (50)	6 (40)	1.03 (0.95–1.11, *p* = 0.47)	0.69 (0.26–1.88, *p* = 0.47)

## Data Availability

Data sharing not applicable. The approved protocol and participant consent does not enable data sharing.

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
