# Peer review of "Participant Choice towards Receiving Potential Additional Findings in an Australian Nephrology Research Genomics Study"

_genes, 2022, doi:10.3390/genes13101804_

Round 1

Reviewer 1 Report

I am happy to accept the manuscript in its current format, it makes for a very interesting read and has the potential to affect the national registry landscape.

Reviewer 2 Report

The article presented by O Shea and colleagues entitled: "Participant choice towards receiving potential additional findings in an Australian nephrology research genomics study" is an original and well-written manuscript regarding an actual subject. Here, there are my comments:

1.- In the abstract and the introduction, please define the abbreviature "IFs", introducing the term.

2.- Regarding this paragraph:......The definition of AFs is “the results of a deliberate search for pathogenic or likely pathogenic variants in genes that are not apparently relevant to a diagnostic indication for which the sequencing test was ordered.”[4].... please correct the orthography`..the dot after the reference and in my opinion,  the authors must discriminate between Target gene sequencing panel and WES Thtah may imply different meanings.

3. At the end of the introduction I will use genomic management instead of

genomic care.

4.- In the result section....In a cohort of 210 participants who underwent research genomic sequencing... please define what type of genomic studies were used in the cohort... WES, TGS.

5.- Second paragraph of Results: it may help to know the statistical test used in each case.

6.-At the end of the result section:  and where the consent process was undertaken by an adult nephrologist (p=0.01)...... this is novel and interesting data, I will expect more discussion.
